# One pot conversion of phenols and anilines to aldehydes and ketones exploiting α gem boryl carbanions

Kanak Kanti Das[1], Debasis Aich[1], Sutapa Dey[2] & Santanu Panda [1]✉

Functional group interconversion is an important asset in organic synthesis. Phenols/anilines being naturally abundant and the carbonyl being the most common in a wide range of bioactive molecules, an efficient conversion is of prime interest. The reported methods require transition metal catalyzed cross coupling which limits its applicability. Here we have described a method for synthesizing various aldehydes and ketones, starting from phenol and protected anilines via $Csp^2$-O/N bond cleavage in a one-pot/stepwise manner. Our synthetic method is found to be compatible with a diverse range of phenols and anilines carrying sensitive functional groups including halides, esters, ketal, hydroxyl, alkenes, and terminal alkynes as well as the substitution on the aryl cores. A short-step synthesis of bioactive molecules and their functionalization have been executed. Starting from BINOL, a photocatalyst has been designed. Here, we have developed a transition metal-free protocol for the conversion of phenols and anilines to aldehydes and ketones.

Functional group interconversion is a beneficial synthetic transformation for everyday organic synthesis[1]. The development of environmentally benign and efficient synthetic methods for functional group interconversion is a central goal of current chemistry research. Phenolic compounds, the most abundant secondary metabolites in plants, are found ubiquitously in nature[2]. Phenolic compounds possess a standard chemical structure comprising an aromatic ring with one or more hydroxyl substituents[3]. Anilines are also cheap and abundant chemicals[4]. Carbonylation reaction represents an essential synthetic transformation for converting various cheap and readily available chemicals into a diverse set of valuable products in our daily lives[5]. Traditionally, carbon monoxide was used as a C1 building block under various transition metal-based catalysts[6]. Among different carbonylation methods developed[7], Pd catalyzed carbonylation reactions have been widely applied to synthesize aldehyde, ester, amide, and other carbonyl derivatives in good yield[8]. In addition to aryl halides the aryl triflates and aryl diazonium salts are well-known coupling partners for oxidative addition with Pd-catalyst (Fig. 1d)[9]. Next CO-coordination, 1,2-migratory insertion followed by subsequent reductive elimination in the presence of various nucleophilic partners provided the corresponding carbonyl compounds. Despite its great success, reactions with carbon monoxide are less common in more complex organic syntheses due to the toxicity and risk associated with high-pressure CO cylinders[10]. Also, there are issues related to pharmaceuticals with permitted daily exposure to transition metals[11]. Hence, some alternative methods also have been developed using photoredox and Ni-chemistry but are constrained by limited scope[12]. Considering that a transition metal-free and CO-free carbonylation of phenols and aniline would be highly useful, but to date no report is available for the transition metal-free deoxygenetive/denitrogenetive conversion of phenol/anilines to aldehydes and ketones hence becoming a topic of prime interest.

Organoboronates have been identified as synthetic anchors owing to their versatile reactivity[13]. Recently, geminal polyboronates have emerged as a distinct class of organoborons for versatile C − C or C −heteroatom bond formations. The easy generation of α-bis(boryl) organometallic species from the corresponding geminal-bis-boron compounds allows diverse synthetic transformations[14]. The presence of empty p-orbital on boron stabilizes the adjacent negative charge. The α-bis(boryl)carbanion species reacts with aldehydes and ketones

[1]Department of Chemistry, Indian Institute of Technology Kharagpur, Kharagpur 721302, India. [2]School of Energy Science & Engineering, Indian Institute of Technology Kharagpur, Kharagpur 721302, India. ✉e-mail: panda.santanu@gmail.com

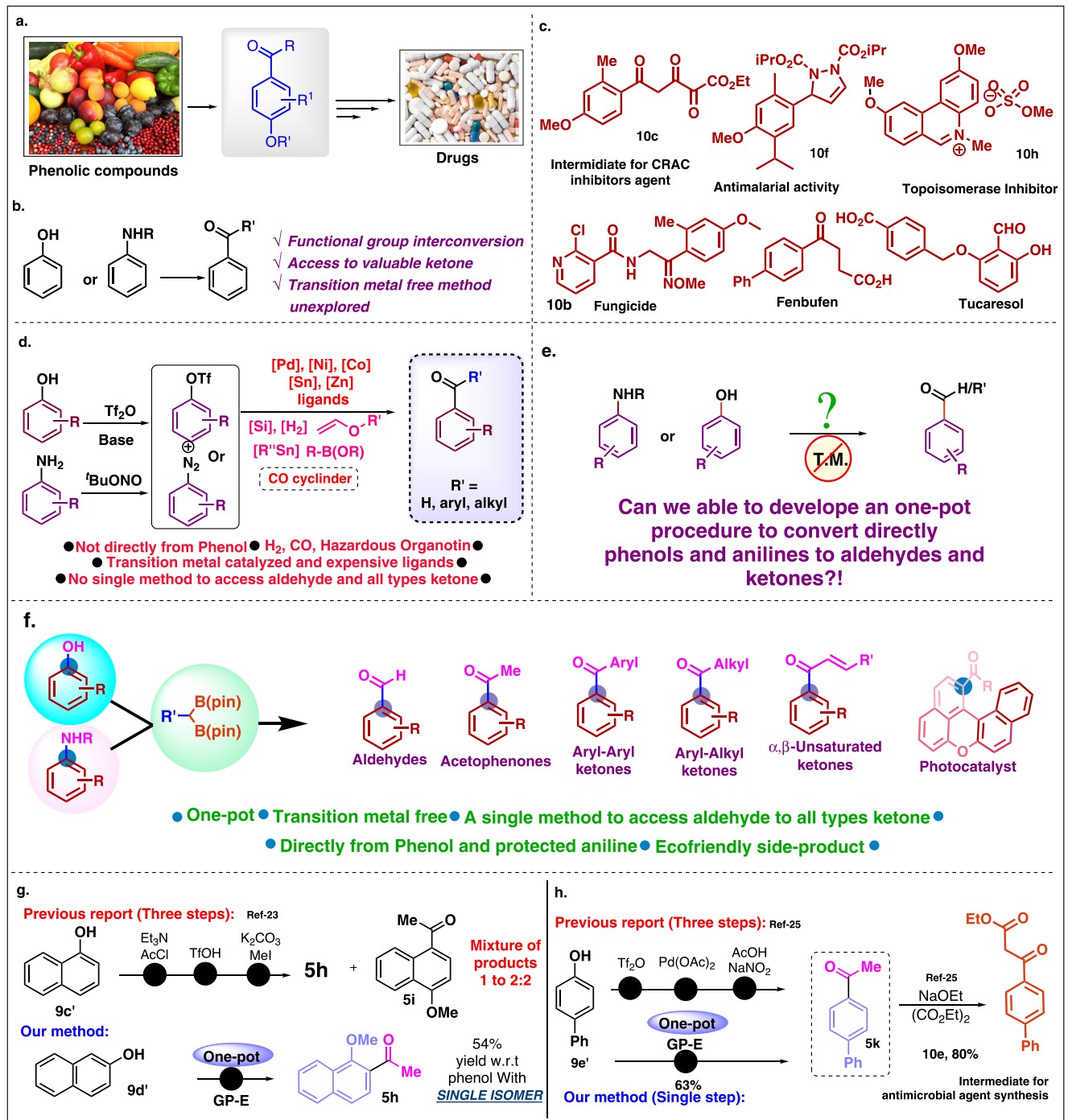

**Fig. 1 | Synthesis of aldehydes, ketones from phenols and anilines and its diverse applications. a** Use of phenols in organic synthesis. **b** Importance of deoxygenartive and deaminative transformations. **c** Bioactive molecules synthesized from aldehydes and ketones. **d** Aldehydes and ketones synthesis using phenols/anilines. **e** Our hypothesis. **f** Our work. **g** Regiospecific synthesis of acetophenone. **h** Short-step synthesis of bioactive molecule.

to yield alkenyl boronates[15]. Matteson established the addition of α-bis(boryl)carbanions to acid chloride, which resulted formation of ketones[16]. Further, this chemistry has been extended to alifatic acids and esters, resulting in alifatic ketones but till to date, it has been never applied for the conversion of phenols/anilines to aldehydes and ketones[17–21]. Therefore, we hypothesized that the α-bis(boryl)carbanions can react with in-situ generated quinketal/imine-ketal for the generation of vinyl boronic esters, and further in-situ oxidation of boronic esters will generate the corresponding aldehydes and ketones. Herein, we developed a methodology for the conversion of both the

phenols and anilines to corresponding aldehydes and ketones in a one-pot manner. The reaction was carried out in a one-pot manner encompassing Csp²-O/N bond cleavage (Fig. 1f). This protocol provides access to the aldehydes and a wide variety of ketones such as acetophenones, aryl-alkyl, aryl-aryl, and conjugated ketones in a quantitative yield directly from easily available phenols and anilines. Further, we have demonstrated the utility of this strategy for the regiospecific synthesis of acetophenones (Fig. 1g), in the short synthesis of bioactive compounds (Fig. 1h) and the direct conversion of bioactive phenol and anilines to the corresponding ketones.

## Results and discussion

With this hypothesis in hand, we have initiated our studies using phenol and homobenzyl geminal boronic ester. To establish the reactivity, we initially decided to isolate the quinketal after the oxidation of phenol using PIDA (phenyliodine(III) diacetate) and then reacted with lithiated geminal-(Bpin) (Supplementary Fig. 1). Next after successive oxidation using $H_2O_2$/NaOH the desired ketone was generated in moderate yield, which justified our hypothesis. We realized that there would be several challenges for conducting this reaction in one pot. We have three major concerns: (1) getting pure quinketal by removing the by-product AcOH coming from PIDA, (2) finding a suitable condition for the 1,2-addition step, and (3) oxidation of intermediate vinyl boronate avoiding competing Dakin oxidation. We initiated our studies using NaOMe as the AcOH quencher which although furnish NaOAc as the byproduct but should not hamper the next step. We have conducted several optimizations to improve the yield (Table 1). We observed that the yield of the 1,2-addition step was highly dependent on the reaction temperature (Table 1, entries 1–3). Further to improve the yield, we screened different bases, in which $K_2CO_3$ was found to be optimal (Table 1, entries 4). Further increase of geminal B(pin) equivalent improved the yield of corresponding ketone up to 80% in a one-pot manner with the formation of phenol from corresponding Dakin oxidation[22] (Table 1, entry 10). A further variation of oxidation conditions from $H_2O_2$ to $NaBO_3$ improved the yield with exclusive formation of the desired product (Table 1, entry 13).

With the optimized conditions in hand, we next sought to determine the generality of the phenol group in this ketone-forming reaction (Fig. 2). First, we have explored the synthesis of various substituted benzaldehydes from the corresponding phenols. Examination of the scope of various phenols revealed that the reaction worked efficiently with various substituents present at the *ortho*, *para*, and *meta*-positions (Fig. 2a–s). Most importantly, this methodology allowed us to introduce a methoxy group at the *para*-position. Starting from monosubstituted phenols, we ended up isolating di-substituted benzaldehydes. Even, by using 2-bromo or 3-chloro phenols, we can isolate 4-methoxy-2-bromo **2 g** or 4-methoxy-3-chloro benzaldehydes **2 h** in great yield. We are surprised to see that phenol-containing free alcohol is equally reactive, affording the desired product in great yield. Next, we have efficiently converted the phenols to corresponding ketones by using substituted geminal boronic esters (Figs. 2, 3a–q). Further, the selective carbonylation for the phenols having free hydroxyl group, ketal, ester, and terminal alkynes is rewarding (Figs. 2, 3o, p, q). Besides, substituted phenols, several fused aryls also have been efficiently acylated with complete regiospecificity. Variation in the germinal B(pin) introduced several branch, acyclic, cyclic, and cyclopropylic ketone (Figs. 2, 4h) in good yields. Interestingly, we have also synthesized the diaryl ketone (**4 g**) by this single method. General methods such as Friedel–Crafts acylation, and Fries rearrangement for the synthesis of acetophenone is always a challenging task as it leads to non-separable regioisomeric products[23]. In-general methods require several steps or transition metal-mediated cross-coupling. Herein, we reported a one-pot method for the regioselective synthesis of acetophenones using methyl geminal B(pin) as the acyl unit (Figs. 2, 5a–e; and see below). Next, we envisioned that the engagement of homoallyl germinal B(pin) would furnish the allyl ketone which might undergo

## Table 1 | Optimization of reaction

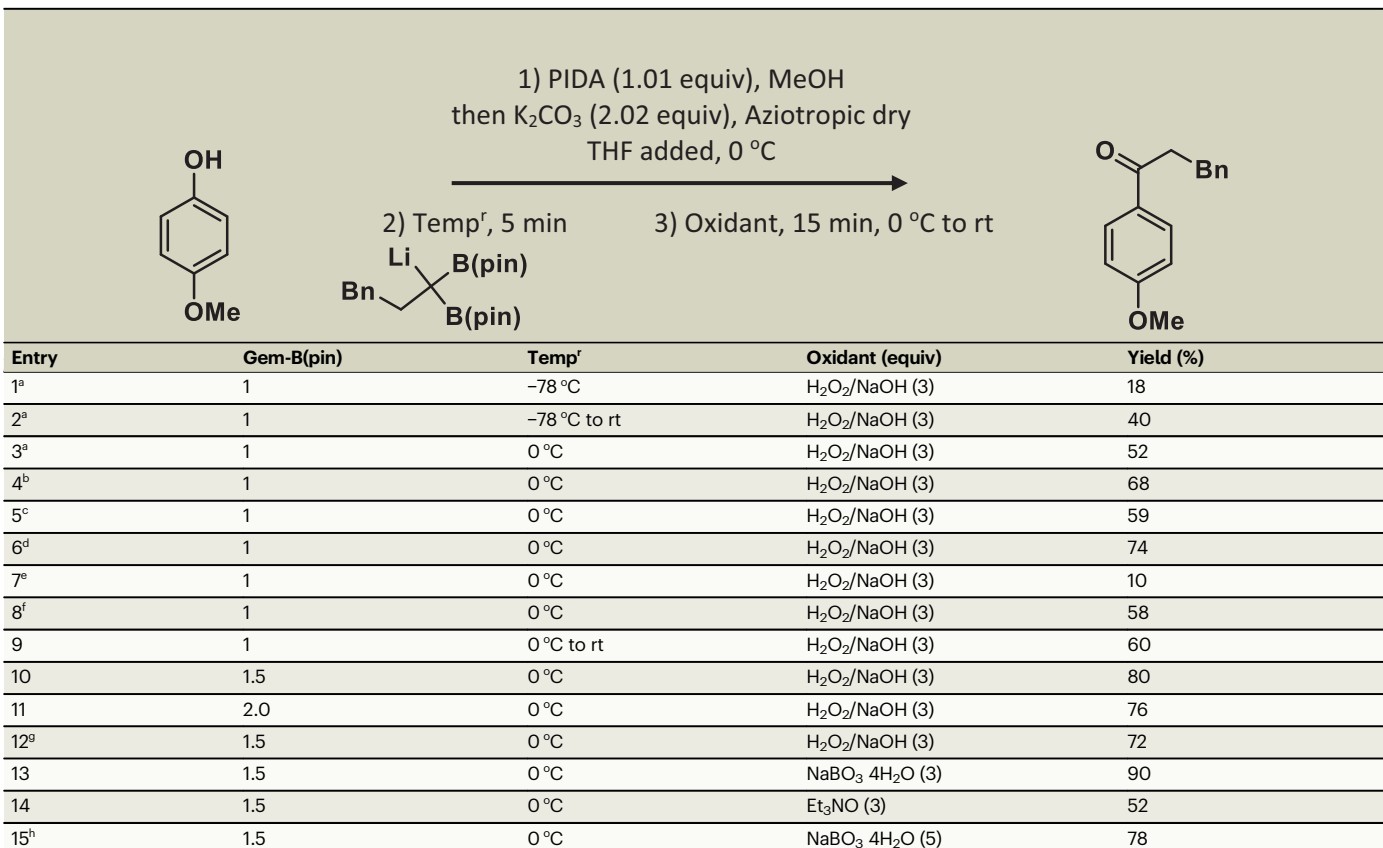

| Entry | Gem-B(pin) | Temp$^r$ | Oxidant (equiv) | Yield (%) |
|---|---|---|---|---|
| 1[a] | 1 | –78 °C | $H_2O_2$/NaOH (3) | 18 |
| 2[a] | 1 | –78 °C to rt | $H_2O_2$/NaOH (3) | 40 |
| 3[a] | 1 | 0 °C | $H_2O_2$/NaOH (3) | 52 |
| 4[b] | 1 | 0 °C | $H_2O_2$/NaOH (3) | 68 |
| 5[c] | 1 | 0 °C | $H_2O_2$/NaOH (3) | 59 |
| 6[d] | 1 | 0 °C | $H_2O_2$/NaOH (3) | 74 |
| 7[e] | 1 | 0 °C | $H_2O_2$/NaOH (3) | 10 |
| 8[f] | 1 | 0 °C | $H_2O_2$/NaOH (3) | 58 |
| 9 | 1 | 0 °C to rt | $H_2O_2$/NaOH (3) | 60 |
| 10 | 1.5 | 0 °C | $H_2O_2$/NaOH (3) | 80 |
| 11 | 2.0 | 0 °C | $H_2O_2$/NaOH (3) | 76 |
| 12[g] | 1.5 | 0 °C | $H_2O_2$/NaOH (3) | 72 |
| 13 | 1.5 | 0 °C | $NaBO_3$ $4H_2O$ (3) | 90 |
| 14 | 1.5 | 0 °C | $Et_3NO$ (3) | 52 |
| 15[h] | 1.5 | 0 °C | $NaBO_3$ $4H_2O$ (5) | 78 |

1 equiv of 4-Methoxyphenol was taken. Dry THF used 1 mL/1 mmol of quinketal. Yields are isolated and calculated from the phenol. [a]NaOMe, [b]$K_2CO_3$, [c]NaHCO_3, [d]$Na_2CO_3$, [e]$Et_3N$ used to quench AcOH. [f]Reaction carried out for 10 min. [g]Oxidation time maintained 30 min. $H_2O_2$/NaOH were premixed at 0 °C before addition. Solid $NaBO_3$ $4H_2O$ was added followed by equal volume water (w.r.t THF) added. [h]Reaction was carried out also with 1 equiv oxidant but result in 56% yield.

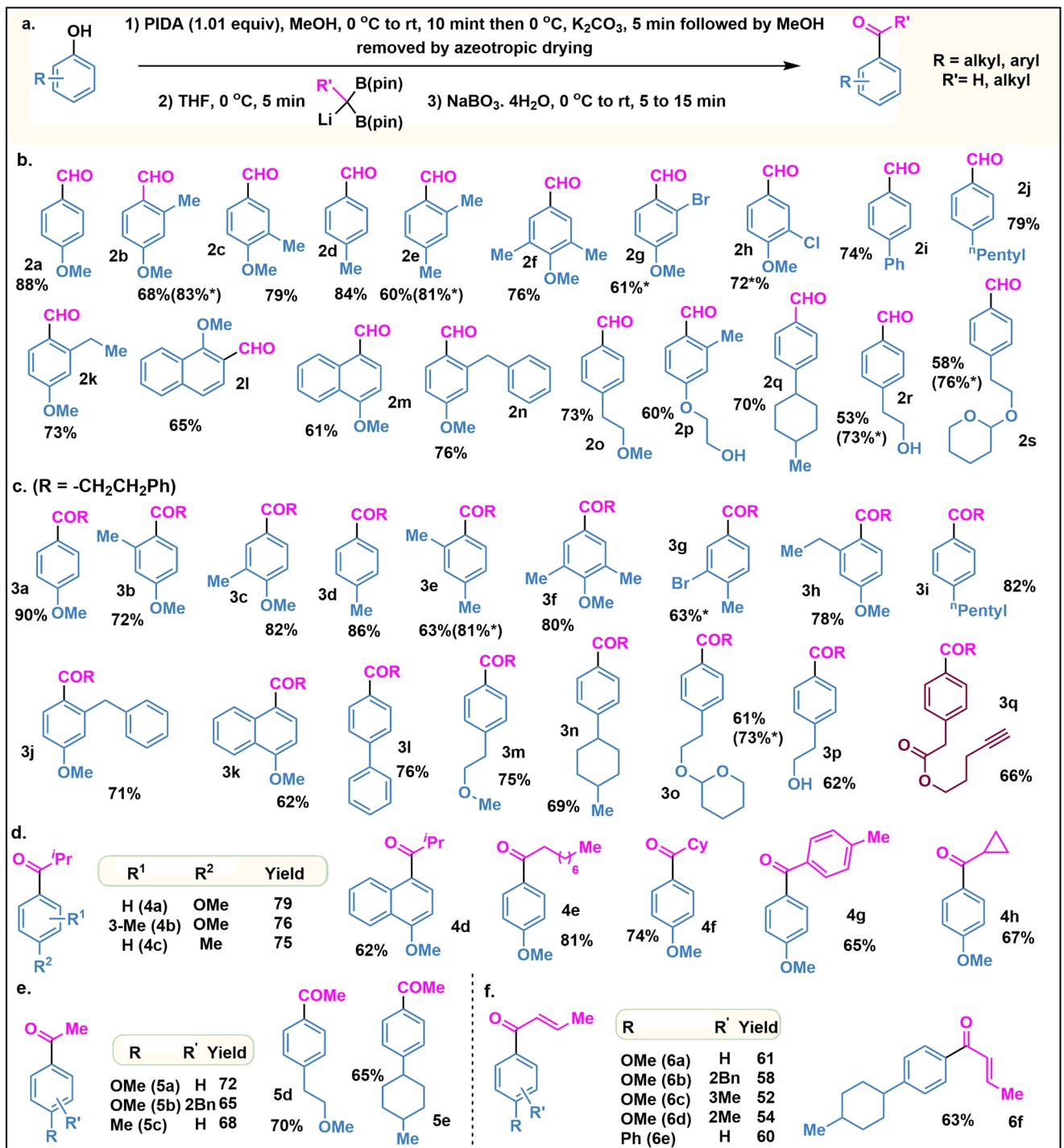

**Fig. 2 | Scope for the aldehydes and ketones directly from phenols. a** Optimized reaction condition. **b** Scope for Aldehydes. **c** Scope for Ketones ($R$ = -CH$_2$CH$_2$Ph). **d** Ketone scope with B(pin) Variation. **e** Scope for the acetophenones. **f** Scope for the α,β- unsaturated ketones. All the reported yields are isolated yields and have been calculated from phenols. The yields with * marks have been calculated from corresponding quinketal.

olefine isomerization and will result in α,β-unsaturated ketones. Gratifyingly, we have synthesized a variety of α,β-unsaturated ketone with 100% *trans* selectivity in good yield (Figs. 2, 6a–f). Anilines are also very cheap and abundant chemicals. We initiated our studies by converting protected aniline to the corresponding imine-ketal[24] core following the reaction with α-bis(boryl)carbanions, but no desired product formation occurs. Further optimization revealed that the isolation in quinketal[25,26] stage is necessary for good yield. A good number of

anilines were converted to the corresponding aldehydes and ketones in good yields (Figs. 3, 7a–o).

Acyl is fairly common in bioactive structures. The group being planar, it can interact with a binding site as an H-bond acceptor through the two lone pairs of electrons on the carbonyl oxygen or via dipole moment. Besides, the acylation of the bioactive phenolic core also leads to APIs. Over the past decade, molecular synthesis has gained extensive impulse by the late-stage functionalization (LSF)

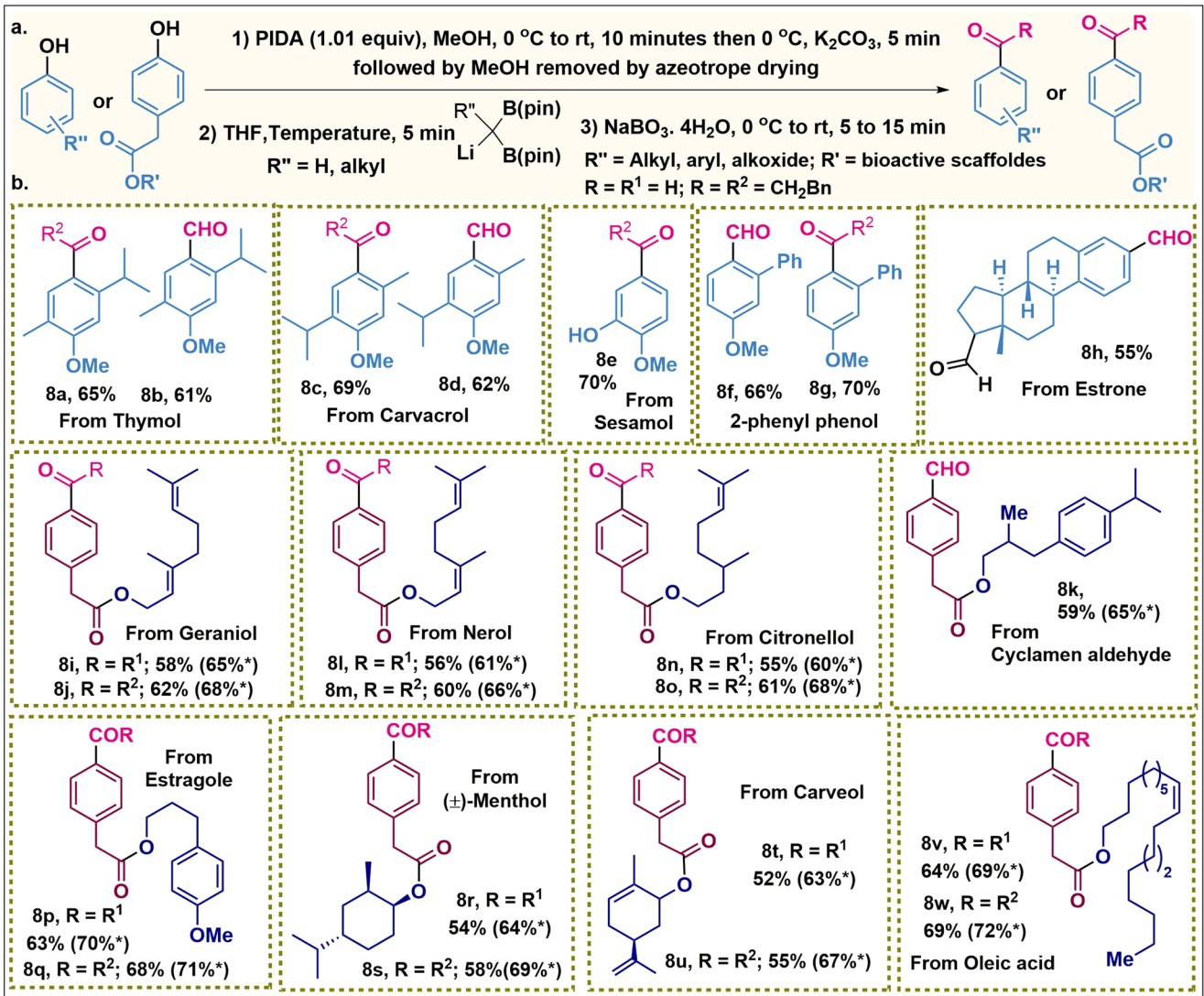

**Fig. 3 | Scope for the aldehydes and ketones starting from anilines. a** Optimized reaction condition. **b** Scope from anilines. All the yields have been calculated from corresponding quinketals.

**Fig. 4 | Bioactive molecules functionalization. a.** Optimized reaction condition. **b** Scope for the bioactive aldehydes and ketones synthesis. All the reported yields are isolated yields and have been calculated from phenols. The yields with * marks have been calculated from corresponding quinketal.

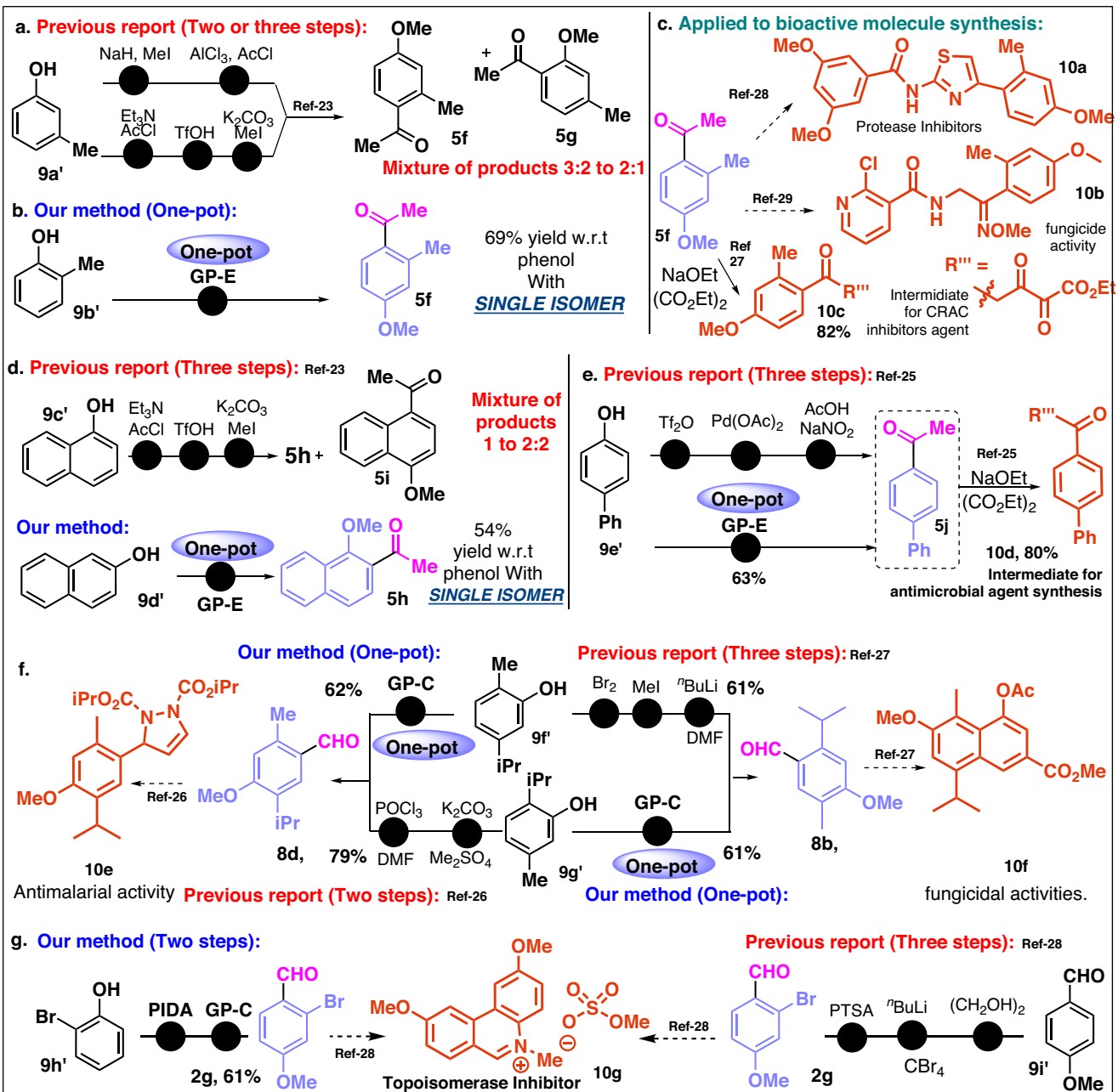

**Fig. 5 | Short step synthesis, regioselective synthesis of acetophenone and application to bioactive molecule synthesis. a** Previous method for 5 f synthesis. **b** Our method for 5 f synthesis. **c** Use of 5 f for bioactive molecule synthesis. **d** Previous method and our method for 5 h synthesis. **e** Previous method and our method for 5j synthesis. f. Previous method and our method for 8b and 8d synthesis. **g** Previous method and our method for 10 g synthesis.

methodologies. This method allowed us to build a technique in the diversification of pharmaceuticals towards medicinal chemists and in drug discovery programs via the controlled installation of small groups at diverse locations. Here we have subjected several bioactive molecules for LSF, which resulted in the desired product with excellent yield as well as high chemoselectivity (Fig. 4). This method was also found to be compatible with ester, olefin, etc, and stereoconvergent in nature. Let's have a look, how efficient out strategy is compared to the reported method. We have elaborated a few examples which demonstrated the applicability of our method compared to the multi-step synthesis of the same material using traditional synthesis. The synthesis of **5 f** from *meta*-cresol requires two or three steps which even resulted in o-,m- acylated product in the ratio of 3.2:2.1 regioisomeric mixture (Fig. 5a)[23]. However by our method, the regiospecific synthesis of **5 f** has been achieved (Fig. 5b)

and further utilized for the synthesis of CRAC inhibitor **10 c**[23,27–29]. Next we resolved the regiospecific synthesis of **5 h** while the reported procedures require three steps resulting with 1:2.2 regioisomeric mixtures (Fig. 5d)[23]. Compared to the general method, the synthesis of **5j** (Fig. 5e) has been demonstrated in one pot manner and further engaged for the synthesis of bioactive compound **10d**[30,31] respectively. The antimalarial[32] compound **10e** and fungicidal[33] **10 f** which are known to synthesize from the aldehydes **8d** and **8b** have been synthesized in one pot manner. Furthermore, the synthon **2 g** for the synthesis of **10g**[34] is reported to be accessed by three steps, while we have reduced it to two steps (Fig. 5g).

While working with the binol system, we hypothesized that a similar reactivity could be explored toward BINOL, and in that case, we might end up with some sort of polycyclic hydrocarbon with the scaffold (Supplementary Fig. 2 and 4). Gratifyingly, we were

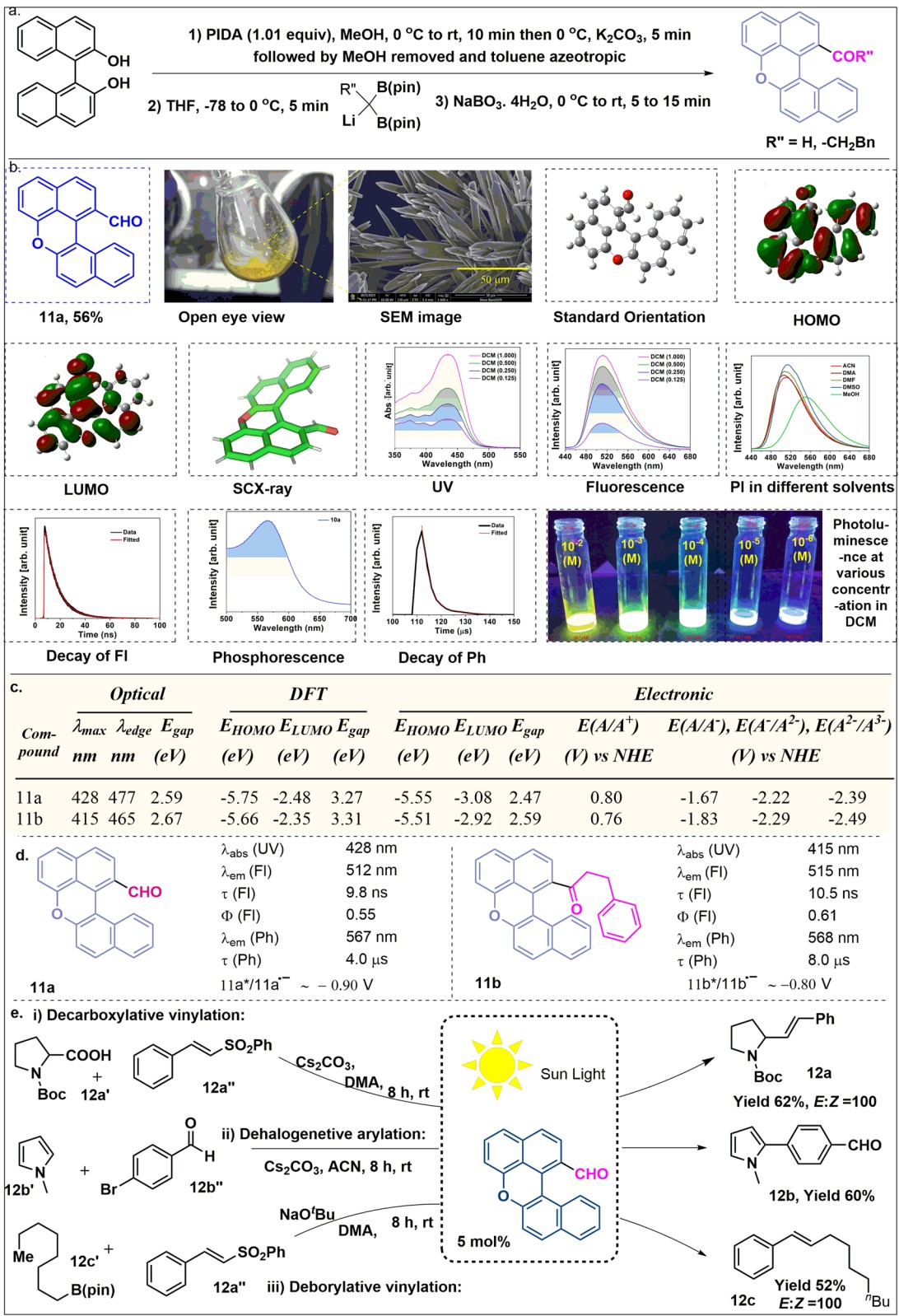

**Fig. 6 | BINOL based photocatalysis; Synthesis, photophysical study and application of 11a as the photocatalyst under sunlight. a** Optimized reaction condition. **b** Photophysical images and SC-XRD of 11a. **c** Numerical data of HOMO-LUMO of 11a and 11b. **d** Photophysical data of 11a and 11b. **e** Reactions using 11a as the photocatalyst. Here Fl means Fluorescence and Ph means Phosphorescence.

surprised to see the formation of compounds **11a** and **11b** while using different germinal B(pin), which is a single step to access functionalizable ODA-analogous[35] and we did not find any kind of photocatalyst having this core. By SEM study we found that both the

compounds have a niddle-like planer shape indicating higher surface area compared to bulk counterparts, which provide higher active sites for catalytic reactions[36]. Their single crystalline nature eliminates the possibility of recombination at the grain boundary related

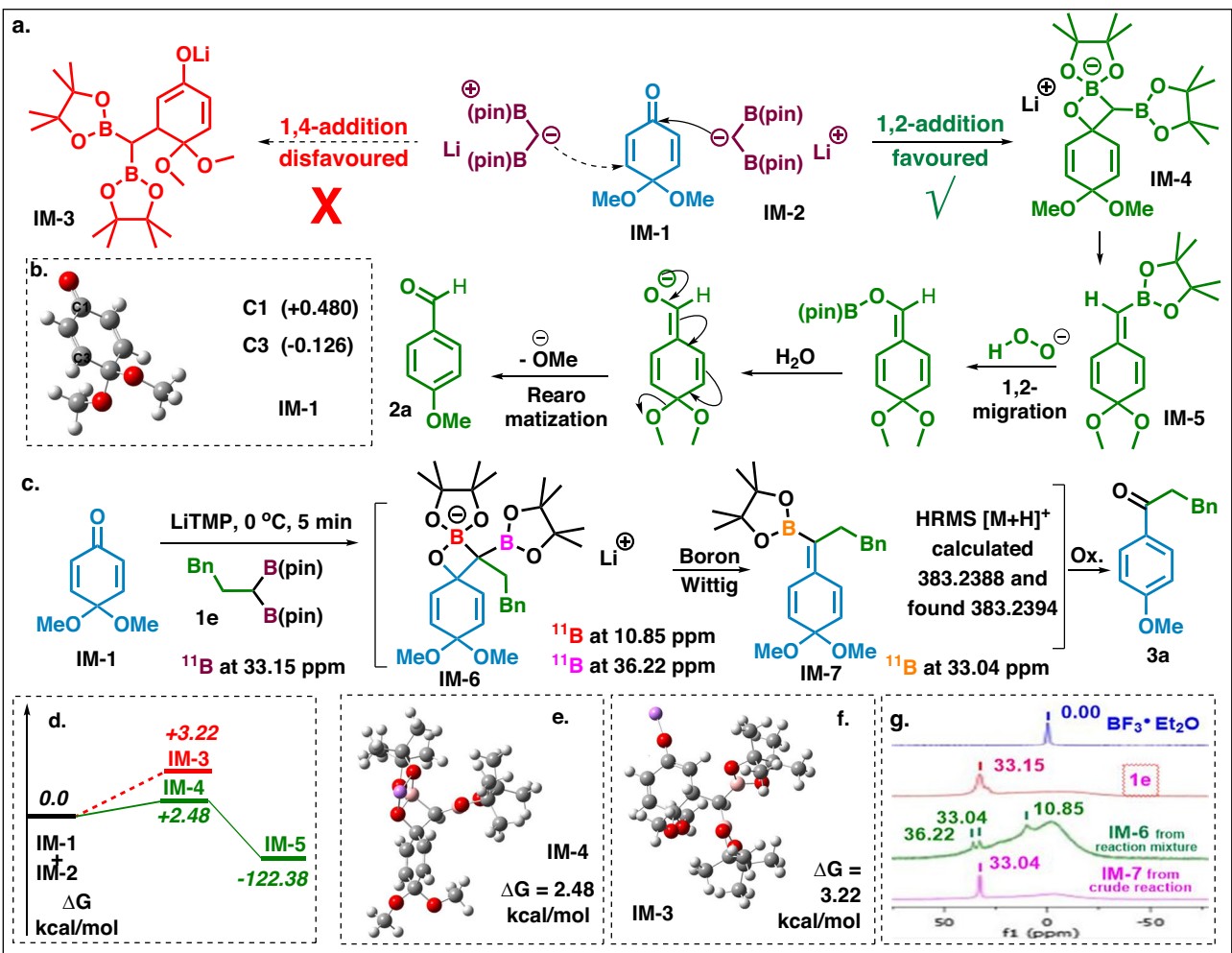

**Fig. 7 | Probable mechanism, DFT calculation and 11B NMR study. a** Proposed mechanism. **b** NBO calculation. **c** [11]B NMR values. **d** Energy difference diagram. **e** Optimized intermediate structures IM-4. **f** Optimized intermediate structures of IM-4. **g** [11]B NMR.

defects zone, exhibiting the feasibility for efficient charge transport[37,38] in solar cell applications (Supplementary Fig. 20). Further from the photophysical study, both of them exhibit fluorescence and phosphorescence emission in visible light. The HOMO-LUMO energy has been computed both by optically and by DFT (Fig. 6c). Cyclic voltammetry (CV) experiments demonstrate that **11a** and **11b** have an oxidation potential ~ 0.9 V and ~ 0.8 V (vs. SCE), respectively, suggesting the possibility of an efficient SET reduction of several organic substrates (Fig. 6d). The photocatalytic activity[39] was elucidated for **11a** under sunlight irradiation (Fig. 6e) which resulted in dehalogenetive arylation as well as decarbobylative and deborylative vinylation with 100% *trans* selectivity[40,41]. Besides their utility as an organic semiconductor and, organic field-effect transistors (OFETs), are under process[42,43].

The mechanistic details of our proposed transformation are outlined in Fig. 7. We have done experimental and theoretical studies to establish the mechanism. Based on the HSAB[44] concept, organolithium being a hard nucleophile, it only undergoes 1,2-addition. But there are some reports available on 1,4-addition using stabilized organlithium[45]. In the case of **IM-2**, the negative charge being stabilized by the boryl group, there might be a possibility for 1,4-addition. But we are getting the1,2-addition product exclusively. The NBO analysis of **IM-1** revealed that **C-1** is more positively charged (+0.480) compared to **C-3** (−0.126), showing that **C-1** is a more favored site for the nucleophilic attack (Supplementary Fig. 23).

Further support came from the DFT calculation suggesting the formation of adduct **IM-4** is exergonic and hence more favored compared to **IM-3** adduct, which demands an additional 0.74 kcal/mol energy (Supplementary Fig. 24). We have also taken the [11]B-NMR, which supports the probable mechanism by observing three different peaks (10.85, 33.04, 36.22) from the crude reaction mixture (Supplementary Fig. 25)[46,47]. The presence of **IM-7** was further confirmed by HRMS data and [11]B-NMR data. We are unable to isolate **IM-7** for further characterization.

In conclusion, we have developed a transition metal-free pioneering strategy for the synthesis of a diverse range of aldehydes and ketones, starting from phenols in a one-pot manner and from protected anilines after the isolation of quinketal via C-O and C-N bond cleavage. A wide range of mono aryl, fused biaryl as well as substituted phenols and anilines have been successfully acylated with good to excellent yield. A variety of ketones, such as aryl-alkyl, aryl-aryl, acetophenones, and conjugated ketones, have been synthesized by this single protocol. This method has an excellent selectivity towards the keto and 1,2-addition over the ester and 1,4-addition. The regioselective synthesis of acetophenones has been developed with 100% selectivity. Several bioactive molecules and pharmaceutically active intermediates have been synthesized within short steps from the reported procedure. Further, we have designed a fused BINOL-based polycyclic compounds having a tuneable end and further exploited as a photocatalyst under sunlight irradiation.

## Methods

To a flame-dried reaction tube the corresponding phenol (1 equiv, 0.5 mmol), was taken and dry MeOH (1 mL/0.5 mmol of phenol) was added. The reaction mixture was cooled to 0 °C and slowly PIDA (1.1 equiv) was added to it. Then the reaction mixture was allowed to stir at rt for 15 min. The reaction mixture was again cooled to 0 °C and to that 2.2 equivalent solid dry potassium carbonate was added. The reaction mixture was allowed to stir for 5 min at the same temperature. Then the MeOH was removed in a vacuum and it was azeotropically dried by dry toluene. Next dry THF was added (2 mL) and cooled to 0 °C. Further, 1.5 equiv of pre-prepared lithiated germinal B(pin) (pre-cooled at 0 °C) was added to it. The reaction mixture was allowed to stir for 5 min at the same temperature. Next 3 equiv of solid $NaBO_3 \cdot 4H_2O$ followed by 2 mL of water were added to it. The reaction mixture was allowed to stir at rt for 5 min. Next EA was added and the aqueous layer was extracted with EtOAc (3×10 mL). Then the solvent was removed under reduced pressure. The resultant crude product was purified by flash chromatography on silica using 10% to 15% EtOAc/Hexane.

## Data availability

The data supporting the results of this work are included in this paper or in the Supplementary Information and are also available upon request from the corresponding author. Crystallographic data for the structures reported in this Article have been deposited at the Cambridge Crystallographic Data Centre, under deposition numbers CCDC 2304923 (11a or 3337), 2304924 (3562 or 11b). Copies of the data can be obtained free of charge via https://www.ccdc.cam.ac.uk/structures/. The coordinates of the optimized structure have been provided in "Source data" file. Source data are provided with this paper.

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

## Acknowledgements

This work was supported by DST(CRG/2020/001892) research grants. K.K.D. wants to thank IIT Kharagpur for the fellowship. D.A. thanks CSIR for his fellowship. S.D. would like to acknowledge the Science & Engineering Research Board (Sanction No. PDF/2022/000706, DST-SERB, India) for providing the National Post-Doctoral Fellowship. The authors thank to DST and SC-XRD LAB, SAIF, IIT Madras for "Single Crystal -X-ray structure solution and refinement". Special thanks to Dr. P. K. Sudhadevi Antharjanam, technical officer, for solving the structures. The authors also thank to Indian Science Technology and Engineering Facilities Map (I-STEM), a Program supported by the Office of the Principal Scientific Adviser to the Govt. of India, for enabling access to the Steady State Photoluminescence (SSPL), funded by Indian Institute of Technology Delhi to carry out this work. Special thanks to Dr. Manjari Chakraborty, Sr. project scientist for helping us record the phosphorescence lifetime.

## Author contributions

S.P. and K.K.D. conceived and designed the experiments. K.K.D. performed the substrate scope experiments. D.A. performed few substrates in Fig. 2d and Fig. 4. S.D. performed the photophysical study in Fig. 6. All authors critically reviewed the manuscript and approved the final version.

## Competing interests

The authors declare no competing interests.
