## [Peer Review File · Nature Communications]

One-pot Conversion of Phenols and Anilines to Aldehydes and Ketones Exploiting α -gem-Boryl CarbanionsREVIEWER COMMENTS

Reviewer #1 (Remarks to the Author):

The manuscript submitted by Panda and co-workers reports an interesting transformation for the direct conversion of phenols and anilines into aldehydes and ketones under transition-metal-free conditions. The reaction proceeds through the α -bis(boryl)carbanions reacting with in-situ generated quinone/imine-ketal intermediates from phenols and anilines in the presence of PIDA to form vinyl boronic esters. Further in-situ oxidation of these boronic esters provides the corresponding aldehydes and ketones. The substrate scope is broad, yielding a range of aldehydes and ketones in moderate to good yields. The authors also demonstrate a short-step synthesis of bioactive molecules, highlighting the advantages of this work. Additionally, the authors present a fused polycyclic-based chromophoric core, designed from BINOL, which functions as an efficient photocatalyst under sunlight irradiation. While the concepts of each step are well-established, the authors cleverly merge these well-known concepts. Therefore, I recommend the acceptance of this work in Nature Communications after addressing the following points.

1) The scheme currently includes an excessive number of colors, resulting in poor readability. I suggest reducing the number of colors used, if necessary, to improve clarity. Check the structures of 9b' and 9h'. The methoxy group is missing.

2) I wonder if the authors can install amide functionality by employing N-substituted diborylmethane (Angew. Chem., Int. Ed. 2022, 61, e202209079).

3) I find many grammatical errors or typos in the manuscript.

Table 1: "mints" needs to be changed to "min"

Table 1: "Aziotrop" needs to be changed to "Azeotrope" or "Azeotropic"

Scheme 1: "krtones" need to be changed to "ketones"

Scheme 2 : "yeilds" needs to be changed to "yields"

Scheme 6 : "dehalodenative" need to be changed to "dehalogenative"

Reviewer #2 (Remarks to the Author):

The author has a novel novel, one-pot, transition metal-free method for synthesizing various aldehydes and ketones, directly starting from phenol and anilines for the first time via Csp² - O/N bond cleavage. This practical method not only could synthesize a diverse range of bioactive molecules, but also construct the BINOL core, which could be used for an efficient photocatalyst under the sunlight light. It seems to be very interesting and practical. Thus, this paper was recommended to be accepted after major revision. In addition, please address the following problem:

1) In Scheme 2, in the research of the synthesis of various benzaldehydes from phenol, the substitution of the phenyl group are all the electron-donating groups including methyl and methoxyl group. Were the electron-withdrawing group substituted phenol tried? Could you tell us the result?

2) The sentence "Further to improve the yield, we screened different bases, in which K₂CO₃ found to be optimal (Table 1, entries 4)" was contradictory to "We did not observe any improvement in yield by varying different bases required to quench the acetic acid." Please modify it.

- 3) "Herein, we reported a one-pot method for the regioselective synthesis of acetophenones using methyl geminal B(pin) as the acyl unit (Scheme 2, 5a-5k)." We could not find the compound 5k in Scheme 2.
- 4) "The synthesis of 5f from meta-cresol requires two or three steps, which even resulted in o-,m- acylated product in the ratio of 3.2:2.1 regioisomeric mixture (Scheme 5a)". Please cite the corresponding literatures to support the sentence.
- 5) "Compared to the general method, the synthesis of 5j and 5k (Scheme 5f) have been demonstrated to in one pot manner and further engaged for the synthesis of bioactive compounds 10d, and 10e". Labeling of compounds seems to be confusion, please check the labling of all the compounds carefully.
- 6) In the research on the synthesis of the benzaldehydes from the protected anilines, no the desired product was obtained in one-pot. The quinketal intermediate seems to be isolated in advance. However, in the abstract and conclusion section, the sentence "A novel, one-pot, transition metal-free method has been described for synthesizing various aldehydes and ketones, directly starting from phenol and anilines for the first time via Csp² -O/N bond cleavage" was described. The author should rewrite them.
- 7) In scheme 5, the author compared the different methods of the compounds 8b and 8d according to the synthetic route length. what are the total yield of different methods?
- 8) To address the generality of the method, the big scale reaction at least 1 g scale should be conducted.
- 9) In this paper, several grammar error existed and should be corrected. For example, B(pin)equivalent should be corrected to Bpin equivalent; "This method allows to build a technique" should be corrected to "This method was allowed to build a technique" etc.

Reviewer #3 (Remarks to the Author):

This manuscript describes a one-pot conversion of phenols and anilines to aldehydes and ketones using α -gem-boryl carbanions. This protocol involves a multi-step process to access the challenging aldehyde and ketone derivatives. The PIDA-mediated quinone imine ketal (QIK) formation from the corresponding protected p-anisidine is a well-known chemistry; considerable work has been reported in the literature. However, the authors revealed several important discoveries, including mechanistic investigation and short-step synthesis of bioactive molecules in practical ways. A reasonably large scope of the reaction is presented, and the application of the present strategy is demonstrated by the synthesis of biologically active molecules and a fused polycyclic-based chromophoric core (photocatalyst). The manuscript is well presented, and the supporting information is good quality. Several significant strategies have been disclosed as possible solutions to the ketone derivative synthesis, including C-H functionalization of simple arenes where an expensive directing group is employed. Importantly, such ketone group installation at the para position has been reported by Maiti and co-workers (<https://www.nature.com/articles/s41467-018-06018-2>; Nat Commun 2018, DOI: 10.1038/s41467-018-06018-2). This important citation is missing. Moreover, another citation is also missing, which involves PIDA-mediated quinone imine ketal formation from the corresponding p-anisidine for the C-C bond formation reaction; Org. Lett. 2023, 25, 32, 6029–6034 (<https://doi.org/10.1021/acs.orglett.3c02181>). Overall, this is a nice strategy for synthesizing various aldehydes and ketones, directly starting from phenol and anilines via Csp² -O/N bond cleavage. Therefore, I would like to recommend this manuscript for publication in this prestigious journal after addressing the following points.

i) You should have two important citations in your reference part that need to be included.

- ii) If any citations are available, you can include those for compound IM-6 and IM-7. So that you can compare the ¹¹B-NMR with the literature.
- iii) You haven't shown any negative experiments in your manuscripts. You can include some failed reactions so that it will be helpful in total synthesis in general.
- iv) Authors have claimed that several sensitive functional groups can be tolerated when converting phenols/amines to aldehydes/ketones. How about testing a few more functional groups, such as amines, nitrile, ketone, amide and anhydrides?
- v) Can you explain more about why the free aldehyde group in Estrone does not react with the more reactive organolithium compound? (compound number 8h, scheme 4)
- vi) In most cases, you have blocked the para position of phenol and aniline derivatives. Can you apply the same strategy for anilines and phenols without blocking the para position?
- vii) A few ¹H-NMR are not clean. For example, compound numbers 8c and 8m.
- viii) There are some minor formatting issues with your reference section. You should modify it.
- ix) Why do you need 3.0 equivalents of oxidant in the second step? Is there any rationale behind this except the high yield?
- x) You can add a reference for synthesising lithiated geminal- (Bpin) species. Is it tough to handle? How about the lifetime of your organolithium species? Can you store it?

Answers to comments by the Reviewer 1

General Comment: The manuscript submitted by Panda and co-workers reports an interesting transformation for the direct conversion of phenols and anilines into aldehydes and ketones under transition-metal-free conditions. The reaction proceeds through the α -bis(boryl)carbanions reacting with in-situ generated quinone/imine-ketal intermediates from phenols and anilines in the presence of PIDA to form vinyl boronic esters. Further in-situ oxidation of these boronic esters provides the corresponding aldehydes and ketones. The substrate scope is broad, yielding a range of aldehydes and ketones in moderate to good yields. The authors also demonstrate a short-step synthesis of bioactive molecules, highlighting the advantages of this work. Additionally, the authors present a fused polycyclic-based chromophoric core, designed from BINOL, which functions as an efficient photocatalyst under sunlight irradiation. While the concepts of each step are well-established, the authors cleverly merge these well-known concepts. Therefore, I recommend the acceptance of this work in Nature Communications after addressing the following points.

Response: Thank you so much for such a thorough review and highly appreciate the comments and suggestions, which will definitely contribute to improving the quality of the publication. Please find below a detailed response to each comment:

Comment: 1 The scheme currently includes an excessive number of colors, resulting in poor readability. I suggest reducing the number of colors used, if necessary, to improve clarity.

Check the structures of 9b' and 9h'. The methoxy group is missing.

Response: 1 Thank you very much for your suggestion. We have removed the colour background from all the schemes.

The structure for both the molecules 9b' and 9h' are correct. We start with a phenol first, after the quinketal formation, two methoxy groups get inserted at the para-position, one of them eliminated leaving one methoxy group at the para-position.

Comment: 2 I wonder if the authors can install amide functionality by employing N-substituted diborylmethane (Angew. Chem., Int. Ed. 2022, 61, e202209079).

Response: 2 Thank you very much for your suggestion. Initially while screening the substrate scope with various germinal B(pin)s, we thought about employing N-substituted diborylmethane, however, we were not able to make it work after several efforts.

Comment: 3 I find many grammatical errors or typos in the manuscript.

Table 1: "mints" needs to be changed to "min"

Table 1: "Aziotrop" needs to be changed to "Azeotrope" or "Azeotropic"

Scheme 1: "krtones" need to be changed to "ketones"

Scheme 2 : "yeilds" needs to be changed to "yields"

Scheme 6 : "dehalodenative" need to be changed to "dehalogenative"

Response: 3 Thank you very much for your suggestion. We have corrected all the grammatical errors throughout the manuscript and highlighted them by yellow colour.

Answers to comments by the Reviewer 2

General Comment: The author has a novel novel, one-pot, transition metal-free method for synthesizing various aldehydes and ketones, directly starting from phenol and anilines for the first time via Csp² -O/N bond cleavage. This practical method not only could synthesize a diverse range of bioactive molecules, but also construct the BINOL core, which could be used for an efficient photocatalyst under the sunlight light. It seems to be very interesting and practical. Thus, this paper was recommended to be accepted after major revision. In addition, please address the following problem:

Response: Thank you so much for such a thorough review and highly appreciate the comments and suggestions, which will definitely contribute to improving the quality of the publication. Please find below a detailed response to each comment:

Comment: 1 In Scheme 2, in the research of the synthesis of various benzaldehydes from phenol, the substitution of the phenyl group are all the electron-donating groups including methyl and methoxyl group. Were the electron-withdrawing group substituted phenol tried? Could you tell us the result?

Response: 1 Thank you very much for your suggestion. We examined the 2-nitro phenol and 3-cyano phenol but the formation of the quinketals was found to be messy without any distinct spot. Hence those phenols failed to undergo the acylation.

Comment: 2 The sentence “Further to improve the yield, we screened different bases, in which K₂CO₃ found to be optimal (Table 1, entries 4)” was contradictory to “We did not observe any improvement in yield by varying different bases required to quench the acetic acid.” Please modify it.

Response: 2 Thank you very much for your suggestion. After careful analysis, we find that the line “We did not observe any improvement in yield by varying different bases required to quench the acetic acid.” is inappropriate. Hence we have removed the line.

Comment: 3 “Herein, we reported a one-pot method for the regioselective synthesis of acetophenones using methyl geminal B(pin) as the acyl unit (Scheme 2, 5a-5k).” We could not find the compound 5k in Scheme 2.

Response: 3 Thank you very much for your suggestion and sorry for that, we have corrected the compound numbering it would be **5e** instead of **5k**; corrected one Scheme 2 (**5a-5e**). We have also shown a few examples in Scheme 5, **5f**, **5j**, **5h**.

Comment: 4 “The synthesis of 5f from meta-cresol requires two or three steps, which even resulted in o-,m- acylated product in the ratio of 3.2:2.1 regioisomeric mixture (Scheme 5a)”. Please cite the corresponding literatures to support the sentence.

Response: 4 Thank you very much for your suggestion. We have cited the reference in the manuscript ref 17.

Comment: 5 “Compared to the general method, the synthesis of 5j and 5k (Scheme 5f) have been demonstrated to in one pot manner and further engaged for the synthesis of bioactive compounds 10d, and 10e”. Labelling of compounds seems to be confusion, please check the labling of all the compounds carefully.

Response: 5 Thank you very much for your suggestion. We have corrected the labeling throughout the manuscript and the supporting information, especially in Scheme 5 and Scheme 6.

Comment: 6 In the research on the synthesis of the benzaldehydes from the protected anilines, no the desired product was obtained in one pot. The quinketal intermediate seems to be isolated in advance. However, in the abstract and conclusion section, the sentence “A novel, one-pot, transition metal-free method has been described for synthesizing various aldehydes and ketones, directly starting from phenol and anilines for the first time via Csp² -O/N bond cleavage” was described. The author should rewrite them.

Response: 6 We appreciate your impactful suggestion. We have rewritten the line in both the abstract and the conclusion section.

Comment: 7 In scheme 5, the author compared the different methods of the compounds 8b and 8d according to the synthetic route length. what are the total yield of different methods?

Response: 7 Thank you very much for your suggestion. I have mentioned the total yield of 8d in the manuscript, Scheme 5. For the compound 8b, in reference 29, we did not find any yield they have reported.

Comment: 8 To address the generality of the method, the big scale reaction at least 1 g scale should be conducted.

Response: 8 Thank you very much for your suggestion. We have conducted the reaction for the synthesis of 2a (with the 8 mmol of 4 methoxy phenol, close to 1 gm) which resulted in the desired product in 79% isolated yield, which was included in Scheme 2a.

Comment: 9 In this paper, several grammar errors existed and should be corrected. For example, B(pin)equivalent should be corrected to Bpin equivalent; “This method allows to build a technique” should be corrected to “This method was allowed to build a technique” etc.

Response: 9 Thank you very much for your suggestion. We have corrected the above-mentioned lines along with a few other lines as highlighted in yellow throughout the manuscript.

Answers to comments by the Reviewer 3

General Comment: This manuscript describes a one-pot conversion of phenols and anilines to aldehydes and ketones using α -gem-boryl carbanions. This protocol involves a multi-step process to access the challenging aldehyde and ketone derivatives. The PIDA-mediated

quinone imine ketal (QIK) formation from the corresponding protected p-anisidine is a well-known chemistry; considerable work has been reported in the literature. However, the authors revealed several important discoveries, including mechanistic investigation and short-step synthesis of bioactive molecules in practical ways. A reasonably large scope of the reaction is presented, and the application of the present strategy is demonstrated by the synthesis of biologically active molecules and a fused polycyclic-based chromophoric core (photocatalyst). The manuscript is well presented, and the supporting information is good quality. Several significant strategies have been disclosed as possible solutions to the ketone derivative synthesis, including C–H functionalization of simple arenes where an expensive directing group is employed. Importantly, such ketone group installation at the para position has been reported by Maiti and co-workers (<https://www.nature.com/articles/s41467-018-06018-2>; Nat Commun 2018, DOI: 10.1038/s41467-018-06018-2). This important citation is missing. Moreover, another citation is also missing, which involves PIDA-mediated quinone imine ketal formation from the corresponding p-anisidine for the C–C bond formation reaction; Org. Lett. 2023, 25, 32, 6029–6034 (<https://doi.org/10.1021/acs.orglett.3c02181>). Overall, this is a nice strategy for synthesizing various aldehydes and ketones, directly starting from phenol and anilines via Csp² -O/N bond cleavage. Therefore, I would like to recommend this manuscript for publication in this prestigious journal after addressing the following points.

Response: Thank you so much for such a thorough review and highly appreciate the comments and suggestions, which will definitely contribute to improving the quality of the publication. Please find below a detailed response to each comment:

Comment: 1 You should have two important citations in your reference part that need to be included.

Response: 1 Thank you very much for your suggestion. We have cited the reference in the manuscript ref 8b, 24c.

Comment: 2 If any citations are available, you can include those for compound IM-6 and IM-7. So that you can compare the ¹¹B-NMR with the literature.

Response: 2 We appreciate for your suggestion. We have not found an exact reference for the boron-Wittig intermediate. In our recent publication using bis-pinacol boronic esters, we also observed ¹¹B NMR peak for two different boron atoms (ref 36).

Comment: 3 You haven't shown any negative experiments in your manuscripts. You can include some failed reactions so that it will be helpful in total synthesis in general.

Response: 3 Thank you very much for your suggestion. We have added the failed experiments in the SI section 16.

Comment: 4 Authors have claimed that several sensitive functional groups can be tolerated when converting phenols/amines to aldehydes/ketones. How about testing a few more functional groups, such as amines, nitrile, ketone, amide and anhydrides?

Response: 4 Thank you very much for your suggestion. We have screened various other functional groups including amine, amide, and cyano but we were not able to get pure product (See SI section 16). In the case of reaction using free ketone or aldehyde, we observed a boron-Wittig reaction with them as well.

Comment: 5 Can you explain more about why the free aldehyde group in Estrone does not react with the more reactive organolithium compound? (compound number 8h, scheme 4)

Response: 5 Thank you very much for your concern. Estrone has a free keto group which also reacted under our optimized condition and furnished the homologated aldehyde group.

Comment: 6 In most cases, you have blocked the para position of phenol and aniline derivatives. Can you apply the same strategy for anilines and phenols without blocking the para position?

Response: 6 Thank you very much for your concern. Here our strategy involves the phenol/Anilines oxidation by PIDA/MeOH in the first step. Therefore while performing the oxidation a methoxy group (for the para-substituted phenol/aniline) or two methoxy groups (for unsubstituted phenol) get inserted. Hence we cannot avoid the methoxy group insertion.

Comment: 7 A few ¹H-NMR are not clean. For example, compound numbers 8c and 8m.

Response: 7 Thank you very much for your suggestion. We have purified the compounds and recorded the NMRs.

Comment: 8 There are some minor formatting issues with your reference section. You should modify it.

Response: 8 Thank you very much for your suggestion. We have corrected the formatting of the reference 29.

Comment: 9 Why do you need 3.0 equivalents of oxidant in the second step? Is there any rationale behind this except the high yield?

Response: 9 We found that with 3.0 equivalents of oxidant, good yield was observed. Lowering the amount of oxidant leads to incomplete oxidation whereas higher loading of the oxidant leads to the over-oxidation of the product. As we are conducting the reaction in one pot there could be unreacted boronic esters, which are also getting oxidized.

Comment: 10 You can add a reference for synthesizing lithiated geminal- (Bpin) species. Is it tough to handle? How about the lifetime of your organolithium species? Can you store it?

Response: 10 Thank you very much for your suggestion. We have cited the references for the synthesis lithiated geminal- (Bpin) species in the manuscript (ref 15c, d). It is easy to handle. We have not stored it, we have used it after synthesizing it from corresponding geminal bis-pinacol boronates using LTMP.

We hope that we are able to clarify all the doubts.

REVIEWERS' COMMENTS

Reviewer #2 (Remarks to the Author):

The author has developed a novel novel, one-pot, transition metal-free method for synthesizing various aldehydes and ketones, directly starting from phenol and anilines for the first time via Csp² -O/N bond cleavage. The author has clarified all the doubts from reviewers. Thus, I recommend the acceptance of this work in Nature Communications

Reviewer #3 (Remarks to the Author):

After carefully reviewing the manuscript, supporting information, and response, I found that the authors have provided extensive and insightful information that has cleared all my doubts and queries. The depth of research and clarity of presentation are impressive, and I believe this manuscript would be an excellent addition to the peer-reviewed journal, Nature Communications. Therefore, I highly recommend that it be considered for publication without further modification.